# Research on Twin Extreme Learning Fault Diagnosis Method Based on Multi-Scale Weighted Permutation Entropy

**DOI:** 10.3390/e24091181

**Published:** 2022-08-24

**Authors:** Xuyi Yuan, Yugang Fan, Chengjiang Zhou, Xiaodong Wang, Guanghui Zhang

**Affiliations:** 1Faculty of Information Engineering and Automation, Kunming University of Science and Technology, Kunming 650500, China; 2School of Information Science and Technology, Yunnan Normal University, Kunming 650500, China

**Keywords:** TELM, fault diagnosis, check valve, MWPE

## Abstract

Due to the complicated engineering operation of the check valve in a high−pressure diaphragm pump, its vibration signal tends to show non−stationary and non−linear characteristics. These leads to difficulty extracting fault features and, hence, a low accuracy for fault diagnosis. It is difficult to extract fault features accurately and reliably using the traditional MPE method, and the ELM model has a low accuracy rate in fault classification. Multi−scale weighted permutation entropy (MWPE) is based on extracting multi−scale fault features and arrangement pattern features, and due to the combination of extracting a sequence of amplitude features, fault features are significantly enhanced, which overcomes the deficiency of the single−scale permutation entropy characterizing the complexity of vibration signals. It establishes the check valve fault diagnosis model from the twin extreme learning machine (TELM). The TELM fault diagnosis model established, based on MWPE, aims to find a pair of non−parallel classification hyperplanes in the equipment state space to improve the model’s applicability. Experiments show that the proposed method effectively extracts the characteristics of the vibration signal, and the fault diagnosis model effectively identifies the fault state of the check valve with an accuracy rate of 97.222%.

## 1. Introduction

As the power equipment of slurry pipeline transportation, the stable and reliable operation of high−pressure diaphragm pumps is the basis for ensuring the safe production of the pipeline transportation system. As a critical component of high−pressure diaphragm pumps, check valves receive damage due to the frequent reciprocating movement. Therefore, it is a vulnerable part, and, often, the point of failure of the pipeline transportation system. Hence, monitoring the operating status of the check valve, and studying its fault diagnosis, is of great significance in improving the production efficiency and production safety of the slurry pipeline.

As it is affected by non−linear factors such as load, friction, and impact, the vibration signal of the check valve often shows non−linear and non−stationary characteristics. Time series analysis and forecasting is generally considered an effective method in data mining. A novel framework was introduced for supporting deep learning in enhancing accurate, efficient, and reliable time series models; it ensures a time series is ‘‘suitable’’ for fitting a deep learning model by performing a series of transformations in order to satisfy the stationarity property [1]. As a method based on non−linear dynamics, permutation entropy (PE) is sensitive to the sudden change and impact of non−stable sequences [2]. Therefore, it can be applied to measure the complexity of the time series in the mechanical dynamics system, and is suitable for the feature extraction of fault signals of mechanical equipment under complex working conditions [3]. However, single−scale permutation entropy has difficulty extracting the complete information of the sudden error and shocks [4]. This paper proposes a multi−scale entropy (MSE) feature extraction method for detecting the complexity and randomness of signals at different scales [5]. Drawing on the idea of MSE, the literature proposes multi−scale permutation entropy (MPE), which improves the capability of noise processing [6]. Taking advantage of MPE’s characteristics in extracting signals, it extracts the non−linear fault characteristics from the bearing vibration signal in different scales, and a novel rolling bearing fault diagnosis method based on MPE and support vector machine (SVM) is proposed. [7]. Avoiding the problem that MPE considers only the permutation information of the time series, the multi−scale weighted permutation entropy (MWPE) method is different from MPE in the sense that it suits better signals, having considerable amplitude information, and also succeeds in accounting for the multiple time scales inherent in financial systems. Compared with MPE, WMPE reduces the standard deviation, which ensures the results are more robust. [8,9]. Combining the advantages of MWPE, the literature quantifies the non−linear characteristics of bearing vibration signals. It establishes a bearing fault diagnosis model based on MWPE and extreme learning machine (ELM) to obtain superior recognition accuracy and efficiency [10].

Support vector machine (SVM), back propagation neural network (BPNN), and ELM are commonly used methods in fault diagnosis. BPNN has good classification accuracy in the case of limited samples, but the training speed is slow and it has structural instability; the classical BPNN algorithm easily plunges into local minimums, low converging speed, etc. SVM avoids falling into the local optimal problem, but the classification performance is limited by the choice of kernel function parameters and its own structural parameters. The reason why ELM can be widely used in the field of fault diagnosis depends on the characteristics of ELM: ELM randomly generates and inputs weights and hidden layer deviations, and can establish fault diagnosis models without iteration or optimization. For the binary classification problem of fault diagnosis, ELM is essentially looking for an optimal classification hyperplane. The method based on hyperplane classification has become more popular in a wide range of applications. However, it is still challenging to achieve a satisfactory performance with only one separate hyperplane for the fault diagnosis of check valves of high−pressure diaphragm pumps.

Extending the applicability of ELM, the literature proposed the twin extreme learning machine (TELM) [11] algorithm. The goal of TELM is to optimize two classification hyperplanes, each of which has the smallest distance from one class and the largest distance as possible with the other class. TELM training optimizes two ELMs simultaneously and obtains two, non−parallel, separating hyperplanes, thereby extending the applicability of ELM. Combining the advantages of TELM, this paper establishes a TELM fault diagnosis model based on multi−scale weighted permutation entropy (MWPE). Featuring the characteristic that the vibration signal of the check valve of the high−pressure diaphragm pump changes under non−stable conditions, this paper analyzes the TELM fault diagnosis model based on MWPE, which presents the non−linear dynamic characteristics of the operating check valve. This model expands the applicability of the ELM, and improves the accuracy of the fault diagnosis of the model.

The innovation points and main contributions of this paper can be summarized as follows:(1)The weighting method of arrangement mode is introduced into MPE, and the vibration signal characteristics of bearings and check valves are extracted by a better MWPE method, and the fault characteristics and signals of mechanical parts are accurately and stably expressed;(2)The biggest innovation in this paper is that the TELM fault diagnosis model is proposed for the first time, which accurately identifies mechanical equipment failures by constructing two ELMs and obtaining a classification hyperplane between them, which has not been reported in previous studies;(3)Combining the MWPE feature extraction method with the TELM diagnostic model for the first time, a fast, effect, and accurate fault diagnosis method for mechanical equipment is proposed. In addition, the effectiveness and innovation of the method were verified by the fault diagnosis of two types of mechanical parts, bearing and check valves.

The rest of this paper is organized as follows: Section 2 introduces the basic theory, including the theory of multi−scale weighted arrangement entropy and the theory of twin limit learning machine. Section 3 introduces the fault diagnosis method and implementation process based on MWPE and TELM. Section 4 gives an experimental analysis of bearing and check valve fault diagnosis. Section 5 presents the experimental discussion and conclusions of this paper.

## 2. Basic Theory

### 2.1. Multiscale Entropy (MSE)

Multi−scale entropy (MSE), proposed by Costa et al. [5], was developed from sample entropy. Unlike sample entropy, which only reflects the characteristics of signals at a single scale, MSE reflects the self−similarity and complexity of signals at different time scales. The value of MSE increases with the higher complexity of the signals. Vibration signals generated by equipment faults concentrate in specific frequency bands. MSE that can reflect the self−similarity and complexity of signals at different time scales can extract the inherent characteristics of vibration signals to judge equipment faults. The calculation process of MSE is as follows:

(1) Use Formula (1) to reconstruct the time series {y(t),t=1,2,…,M} into coarse−grained time series {u(t),t=1,2,…,N}, where τ is the scale factor, and the length of the reconstructed time series is N=int(M/τ).
(1)$$u(i)=\frac{1}{\tau }\sum_{i=(j-1)\tau +1}^{i\tau }y(t)$$

Vector x(i) can be calculated by
(2)x(i)=[u(i),u(i+1),…,u(i+m−1)]
where m is the embedding dimension and r is the similar capacity.

Given (i=1−N−m+1), according to Formula (2), the distance between the vector x(i) and other vectors x(j)(1≤j≤N−m+1,j≠i) calculates as:(3)$$d[x(i),x(j)]=\underset{k=0\sim m-1}{max}|u(i+k)-u(j+k)|$$

According to r(r>0), the given similar capacity and value i(1≤i≤N−m+1), calculate the number d[x(i),x(j)]<r, and then calculate the ratio of the number d[x(i),x(j)]<r to the number of vectors N−m according to Formula (4): (4)$$C_{i}^{m}(r)=\{count(d[x(i),x(j)<r])\}/(N-m)$$

Then, calculate the average value $\phi^{m}(r)$ of the results according to Formula (5): (5)$$\phi^{m}(r)=\frac{1}{N-m+1}\sum_{i=1}^{N-m+1}C_{i}^{m}(r)$$

Repeat the above steps, let m=m+1, calculate: $\phi^{m+1}(r)$:
(6)$$\phi^{m+1}(r)=\frac{1}{N-m}\sum_{i=1}^{N-m}C_{i}^{m+1}(r)$$

Obtain sample entropy of coarse−grained time series {u(t),t=1,2,…,N} as: (7)$$A_{p}^{En}(m,r,N)=-In\frac{\phi_{}^{m+1}(r)}{\phi_{}^{m}(r)}$$

Repeat all the above steps to calculate the sample entropy under different values, and obtain the final multi−scale entropy τ.

The value of MSE is related to the embedding dimension m, similarity tolerance r, scale factor τ, and coarse−grained data length N.

### 2.2. Multi−Scale Permutation Entropy (MPE)

The multi−scale permutation entropy (MPE) algorithm was first proposed by Aziz et al. [12]. It is the combination of a multi−scale algorithm and permutation entropy algorithm. The disadvantage of permutation entropy is that the extraction of the sorting mode of each time series does not include other information of the signal, but the sorting mode itself. That is, the process loses the data of amplitude.

The coarse−grained processing of the original time series $\left\{x_{1},x_{2},\ldots ,x_{N}\right\}$ by MPE is the core of the multi−scale algorithm. The steps to construct the multi−scale time series $\left\{y_{j}^{\left(s\right)}\right\}$ are as follows:

(1) Calculate the average value of time series $\left\{x_{i}\right\}$ in each window s according to Formula (8).
(8)$$y_{j}^{\left(s\right)}=\frac{1}{s}\sum_{i=(j-1)s+1}^{js}x(i),\;1\leq j\leq \frac{N}{s}$$

The signal x(i):1≤i≤N is coarse−grained to obtain the coarse−grained sequence $y_{j}^{\left(s\right)}$, $s=1,2,\cdots ,s_{m}$ is the scale factor, and N is the length of the original time series; 

(2) Calculate the permutation entropy (*PE*) of each coarse−grained sequence $y_{j}^{\left(s\right)}$, and while the scale factor is s, the *MPE* is as follows: (9)$$MPE(x,m,\tau ,s)=PE(y_{j}^{\left(s\right)},m,\tau )$$

*MPE* calculates the permutation entropy of the coarse−grained time series, whose core is the coarse−grained processing. When exacting *MPE* features, embedding dimension m, delay time τ, and scale factor s have a great influence on the result. 

### 2.3. Multi−Scale Weighted Permutation Entropy (MWPE)

Multi−scale weighted permutation entropy (MWPE) combines multi−scale analysis with WPE [9]. Multi−scale analysis obtains the time series of original time series at multiple scales through the coarse−grained process, and obtains the complexity of the time signal at different scales; that is, MWPE describes the structural characteristics and complexity of time series at multiple scales. The MWPE calculation steps are as follows:

(1) Obtain the coarse−grained time series on multiple scales of the original time series through coarse−grained process. Time series $\left\{x_{1},x_{2},\ldots ,x_{N}\right\}$ are divided into non−overlapping windows of length s, and calculate the coarse−grained time series $y_{j}^{\left(s\right)}$ on different scale factors s.
(10)$$y_{j}^{\left(s\right)}=\frac{1}{s}\sum_{i=(j-1)s+1}^{js}x(i),\;1\leq j\leq \frac{N}{s}$$

(2) Calculate the coarse−grained sequences $y_{j}^{\left(s\right)}$ at each scale, according to Formula (10) to obtain *MWPE*.
(11)$$MWPE(x,s,m,\tau )=WPE(y_{j}^{\left(s\right)},m,\tau )$$

According to the formula, it can be concluded that when s=1, *MWPE* equals *WPE*. However, for most signals in reality, single−scale weighted permutation entropy cannot wholly describe signals’ internal structural characteristics and complexity. At the same time, *MWPE* can reflect the actual attributes of signals more comprehensively [13]. 

### 2.4. Twin Extreme Learning Machine

The standard ELM [13] comprises three layers: the input layer, the hidden layer, and the output layer. It is developed on the basis of the single−hidden−layer feedforward neural networks (SLFNs), but the hidden layer of SLFN is only a BP neural network with one layer. The topology of ELM is shown in Figure 1.

As an efficient single−hidden−layer feedforward neural network, we assume that for a given n training sample, $X=(x1,\;x2,\;.\;.\;.\;,\;xn)\in R_{}^{d1\times n}$, whose labels are $Y=(y1,\;y2,\;.\;.\;.\;,\;yn)\in R_{}^{n\times d2}$, where d1 and d2 represent the dimensions of the input data and output data, respectively. The weight $W=\omega ij\in R_{}^{d1\times L}$ of the hidden layer of the extreme learning machine is randomly selected, where L represents the number of neurons in the hidden layer. The calculation of the hidden layer is the same as the calculation of the traditional forward propagation network H=g(X,W), where $H\in R_{}^{n\times L}$, and g(·) is the activation function. 

The learning objective of the extreme learning machine is to solve the output weight β by minimizing the sum of the prediction error loss functions. The objective function is: (12)$${\mathrm{minL}}_{\mathrm{ELM}}=\frac{1}{2}||\beta |{|}^{2}+\frac{C}{2}||Y-H\beta |{|}^{2}$$
where the *C* value directly affects the generalization performance of ELM, and is a regularization coefficient.
(13)$$H={\left[\begin{array}{ccc} g(w_{1},b_{1},x_{1}) & \cdots & g(w_{L},b_{L},x_{1})\\ \vdots & & \vdots \\ g(w_{1},b_{1},x_{N}) & \cdots & g(w_{L},b_{L},x_{N}) \end{array}\right]}_{N\times L}$$
(14)$$\beta ={\left[\beta_{1}^{T},\beta_{2}^{T}\ldots ,\beta_{L}^{T}\right]}_{L\times m}^{T}\;$$
(15)$$Y={\left[y_{1}^{T},y_{2}^{T}\ldots ,y_{L}^{T}\right]}_{N\times m}^{T}\;$$

Taking the derivative β of Formula (12) and setting it to 0, the output weight β can be obtained as follows: (16)$$\beta =\left\{{}_{H^{T}{\left(\frac{I}{C}+HH^{T}\right)}^{-1}Y,N<L}^{{\left(\frac{I}{C}+H^{T}H\right)}^{-1}H^{T}Y,N\geq L}\right.$$
where *I* is the identity matrix.

TELM, as an improvement of the extreme learning machine algorithm [11,14], is also a binary classifier. However, it uses two non−parallel hyperplanes instead of one single hyperplane for classification. The TELM works by obtaining two non−parallel hyperplanes by solving two more minor QPP problems. It is assumed that U and V are the data matrices representing the labels of class 1 and −1 outputted after the nodes of the hidden layer, respectively. For the TELM algorithm, it solves two non−parallel hyperplanes in R:(17)$$\begin{array}{l} f_{1}(x):=\beta_{1}\cdot h(x)=0,\\ f_{2}(x):=\beta_{2}\cdot h(x)=0, \end{array}$$

Keeping each hyperplane close to its class’s data and away from other class’s data points.

Then, obtain new data points from the +1 class or −1 class based on these two hyperplanes. The core of the TELM algorithm is to solve the relationship between the following two QPPs objective functions and corresponding constraints:
(18)$$\begin{array}{l} \underset{\beta 1,\xi }{\min}\frac{1}{2}||U\beta_{1}|{|}_{2}^{2}+c_{1}e_{2}^{T}\xi \\ \begin{array}{cc} -V\beta_{1}+\xi \geq e_{2}, & \xi \geq 0 \end{array} \end{array}$$
then
(19)$$\begin{array}{l} \underset{\beta_{2},\eta }{\min}\frac{1}{2}||V\beta_{2}|{|}_{2}^{2}+c_{2}e_{1}^{T}\eta \\ \begin{array}{cc} -U\beta_{2}+\eta \geq e_{1}, & \eta \geq 0 \end{array} \end{array}$$
where ξ and η are the error vectors corresponding to class −1 and class +1 in the training model, respectively; $c_{1},c_{2}>0$ are the trade−offs parameters; and $e_{1}$ and $e_{2}$ are each corresponding vector. 

The dual problem of the original problem of Formulas (18) and (19) is as follows:(20)$$\begin{array}{l} \underset{\alpha }{\max}e_{2}^{T}\partial -\frac{1}{2}\partial^{T}V{\left(U^{T}U+\varepsilon I\right)}^{-1}V^{T}\partial \\ 0\leq \partial_{i}^{}\leq c_{1},i=1,2,\cdots ,m_{2} \end{array}$$
(21)$$\begin{array}{l} \underset{\gamma }{\max}e_{2}^{T}\gamma -\frac{1}{2}\gamma^{T}U{\left(V^{T}V+\varepsilon I\right)}^{-1}U^{T}\gamma \\ 0\leq \gamma_{i}^{}\leq c_{2},i=1,2,\cdots ,m_{2} \end{array}$$

By these two formulas, the optimal sum of Lagrangian multipliers ∂ and γ can be obtained, and the sum of decision variables $\beta_{1}$ and $\beta_{2}$ can be calculated as follows: (22)$$\beta_{1}=-{\left(U^{T}U+\varepsilon I\right)}^{-1}V^{T}\partial$$
(23)$$\beta_{2}=-{\left(V^{T}V+\varepsilon I\right)}^{-1}U^{T}\gamma \;$$

A new data point belonging to $x\in R^{n}$ is assigned to (r=+1,−1) from the class r,
(24)$$f(x)=\arg(\underset{r=1,2}{\min}dr(x))=\arg(\underset{r=1,2}{\min}\left|\beta_{r}^{T}h(x)\right|)$$

## 3. Feature Extraction and TELM Fault Diagnosis Method

Rolling bearings and diaphragm pump check valves are mechanical equipment with complex structures, and the vibration signal excited by the fault location has complex, non−linear characteristics. In addition, as it is affected by the background noise and the acquisition error of the vibration sensor, plus the change in the operating conditions, the acquired vibration signal has serious non−stationary characteristics. In summary, the feature extraction and fault diagnosis of mechanical equipment vibration signals in the actual operating environment face great challenges. The above analysis shows that the multi−scale weighted permutation entropy can effectively retain the coarse−grained information and the amplitude information of the vibration signal, and the twin extreme learning machine can distinguish the two types of samples to the greatest extent through the distance between the two ELM hyperplanes. The fault diagnosis model based on MWPE and TELM effectively makes up for the shortcomings of existing fault diagnosis methods, so it inspired us to propose a new fault diagnosis method. Through the proposed fault diagnosis method, the fault feature extraction efficiency and fault identification accuracy of the bearing and check valve will be further improved.

The specific steps of the MWPE–TELM fault diagnosis method proposed in this paper are as follows:

Step 1: Collect vibration signals from various states of parts;

Step 2: Divide the collected vibration signal data, divide the non−overlapping samples into 60 segments, and extract fault features for each part;

Step 3: For each state, calculate the MWPE features of the vibration signal sample to construct a fault diagnosis feature space, the size of whose feature matrix is 60 × 20, where the scale factor *s* of the MWPE is 20, the embedding dimension *m* is 4, and the time delay γ is 1. The scale factor *s* of MSE and MPE is 20;

Step 4: Input the obtained high−dimensional features into the ELM model for training and testing, 60% as training samples and 40% as test samples;

Step 5: Adopt a mixed−domain ELM fault diagnosis model to identify the fault information.

The algorithm flow chart is shown in Figure 2.

## 4. Experimental Simulation and Analysis

### 4.1. Analysis of Data

This section uses bearing data from Case Western Reserve University in the United States for fault diagnosis to verify the effectiveness of the method proposed in this paper, and the effectiveness of the hybrid domain feature extraction [15].

This paper takes the vibration signal of the fan terminal bearing at the motor speed of 1797 r/min as experimental data, as Table 1 below shows: 

The three fault diameters of the inner ring, outer ring, and rotary body are 0.07 ft, 0.014 ft, and 0.021 ft, respectively, and the Figure 3 below shows their time−domain waveform.

As shown in the figure, signals such as IR007, IR014, IR021, OR007, OR014, and OR021 demonstrate periodic impact, while the signals of B007, B014, and B021 show no obvious periodic hint. The amplitude difference between the signals is not apparent, and neither are the characteristics of the impact or its period. Hence, the proposed mixed domain can identify the bearing fault type and the degree of fault. 

According to the entropy feature extraction method proposed in this paper, divide the vibration signals of each state into 60 non−overlapping segments with a length of 1280.

Since TELM diagnosis results are affected by different entropy features, to eliminate contingency, the above entropy feature extraction method is used to conduct bearing fault diagnosis experiments under other values for the parameter s, and Figure 4 shows the results below, Table 2 shows the optimum value of the diagnosis results under different entropy characteristics.

When extracting the MWPE features, the hidden layer nodes of TELM change from 0 to 2000, and the average fault diagnosis accuracy is greater than 90%. When the hidden layer node takes the value H = 50, the fault diagnosis accuracy is 97.222%. The results in Table 3 show that adopting the proposed MWPE feature extraction method can improve fault diagnosis accuracy, which is more optimal than the MSE and MPE feature extraction methods.

By comparing the four classification models of BPNN, SVM, ELM, and TELM, the bearing fault diagnosis results under different feature dimensions are shown in Figure 5. It can be seen that the TELM method achieves better results under different feature dimensions, with an average accuracy higher than 95%.

### 4.2. Data Analysis of Diaphragm Pump Check Valve

The above experiments show that the proposed MWPE feature extraction method can achieve satisfactory results in bearing fault diagnosis. Thus, it is applied to the fault diagnosis of the diaphragm pump check valve. Figure 6a,b show the sensors that are fixed on the shells of the inlet and outlet valves. For each valve, there is one acceleration sensor of type PCB352C33 (sensitivity 100 mV/g) and one sound pressure sensors of type MP021 (50 mV/Pa). The acceleration sensor collects the shell vibration signal along the Z-axis using three channels, while the sound pressure sensor collects the sound signal along the Y-axis direction. Figure 6c shows the vibration signal acquisition device of the check valve. The eight channel analog signal is amplified, filtered, and converted into A/D by the data acquisition card, and sent to the PS PXI−3050EXT 2.7ghz controller. Then, the signal is transferred to the PS PXIE−9108Ext eight slot industrial computer and stored in the hard disk.

The operation regularly ran from November 1st until the 24th, when the check valve failed and was replaced. The No. 2 feeding check valve of the diaphragm pump was stuck. On 24 December, the No. 3 discharge check valve of the diaphragm pump was worn and broken down.

We randomly selected a group of vibration signals in the time−domain of the normal state of the check valve, the stuck valve fault, and worn valve seat periods. Figure 7 shows the result.

There are some fault shocks in the time−domain diagrams, yet all the local waveforms contain noise, making the shock period unclear. It is difficult to analyze the types and causes of the faults through the time−domain waveforms only. Therefore, we use a vibration signal sample and entropy feature extraction to diagnose the fault of the check valve.

Firstly, we divide the vibration signals of check valves in each state into 60 non−overlapping segments, and the data points of each piece are 1280.

The above entropy feature extraction method is used to conduct check valve fault diagnosis experiments under other values for the parameter s, and Figure 8 shows the results below. Since the fault diagnosis result of the twin extreme learning machine algorithm is affected by the number of hidden layer nodes, Table 4 compares the ELM’s fault diagnosis results of the check valve when the hidden layer nodes range from 0 to 2000.

According to Figure 9, no matter how much the number of hidden layer nodes changes, the average diagnosis result obtained by the check valve MWPE feature is close to 95%, with the maximum diagnosis accuracy being 97.22%. Therefore, the MWPE–TELM feature extraction method proposed in this paper achieves better results in the fault diagnosis of bearings and check valves, proving the method’s effectiveness.

## 5. Conclusions

This paper proposes a fault diagnosis method based on MWPE and TELM, and applies it to the fault diagnosis of the check valve of a high−pressure diaphragm pump. While extracting the arrangement information of the vibration signal, the method also adds up information of amplitude, and uses MWPE to characterize the operating state of the check valve. MWPE can effectively extract the non−linear dynamic characteristics of the check valve of the high−pressure diaphragm pump and capture the sudden change and impact information in the signal. The TELM fault diagnosis model, which trains and optimizes two ELMs simultaneously, establishes two non−parallel classification hyperplanes in the running state feature space, and expands the applicability of ELMs. When the method proposed by this paper is applied to the multiple failures of a check valve, the experimental results show that the proposed algorithm accurately and effectively extracts the failure information of the one−way valve, with a diagnostic accuracy of over 97%. Though the proposed method provides superior accuracy of feature extraction and robustness of the fault diagnosis, the stability of MWPE under coarse−grained data could be further improved, and further research could continue to explore the optimal parameters of TELM.

## Figures and Tables

**Figure 1 entropy-24-01181-f001:**
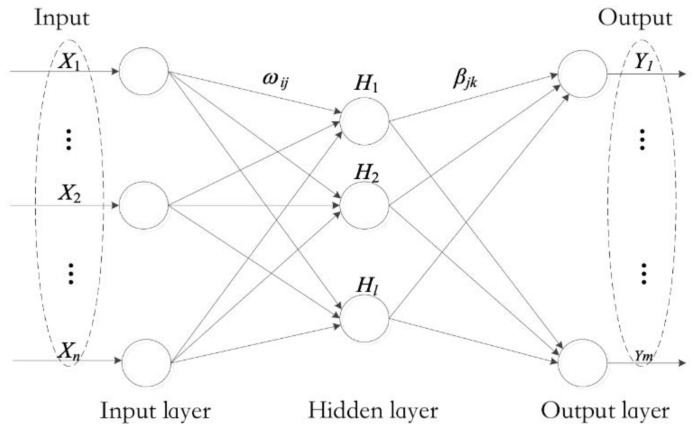
ELM model.

**Figure 2 entropy-24-01181-f002:**
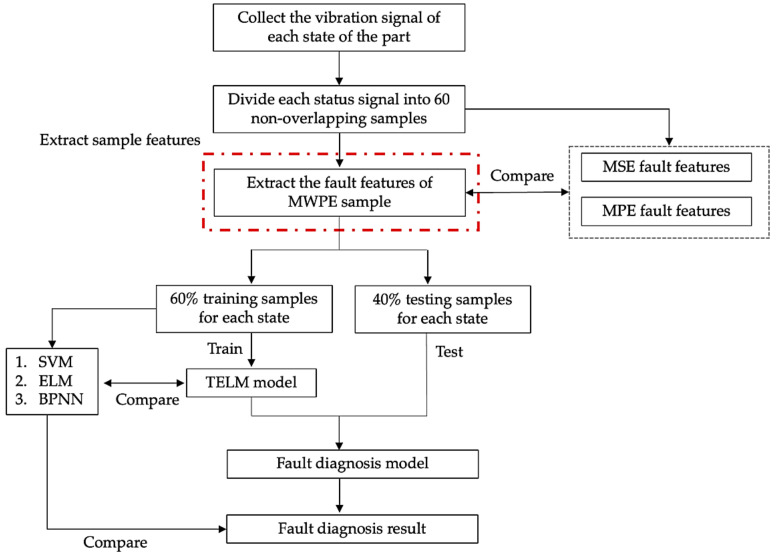
Algorithm flow chart.

**Figure 3 entropy-24-01181-f003:**
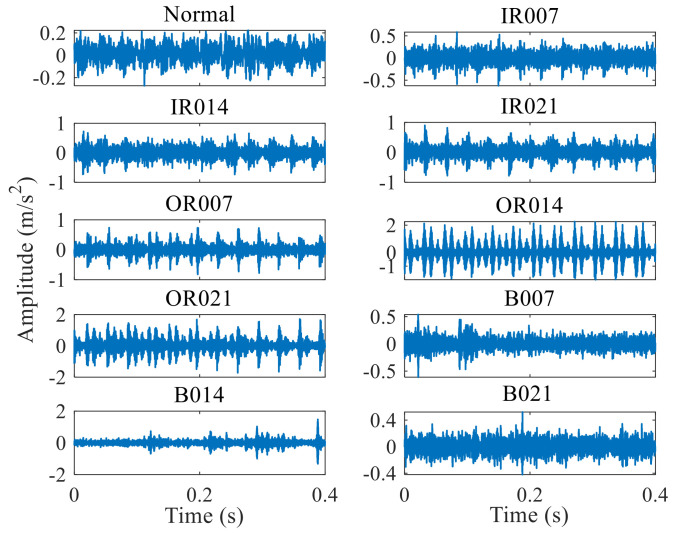
10 bearing states in time−domain Figure 1.

**Figure 4 entropy-24-01181-f004:**
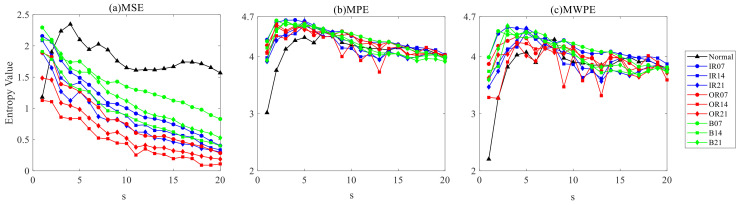
Diagnosis results of MWPE characteristics of bearing under different s values of TELM.

**Figure 5 entropy-24-01181-f005:**
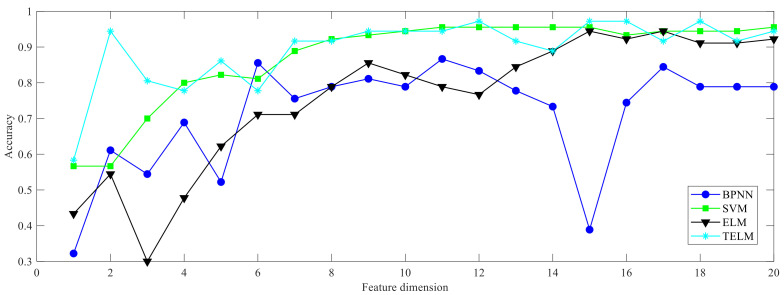
Bearing fault diagnosis accuracy obtained under different classification models.

**Figure 6 entropy-24-01181-f006:**
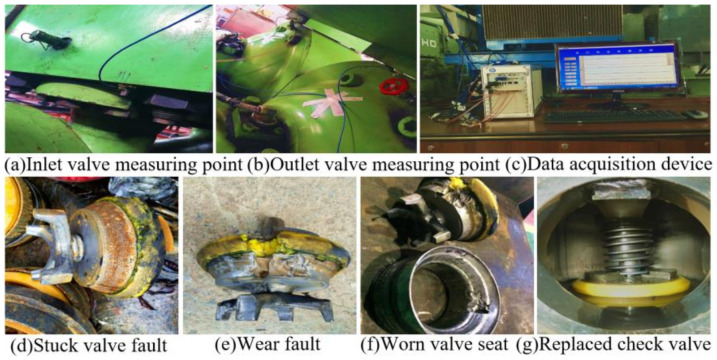
Vibration signal acquisition device and fault check valve.

**Figure 7 entropy-24-01181-f007:**
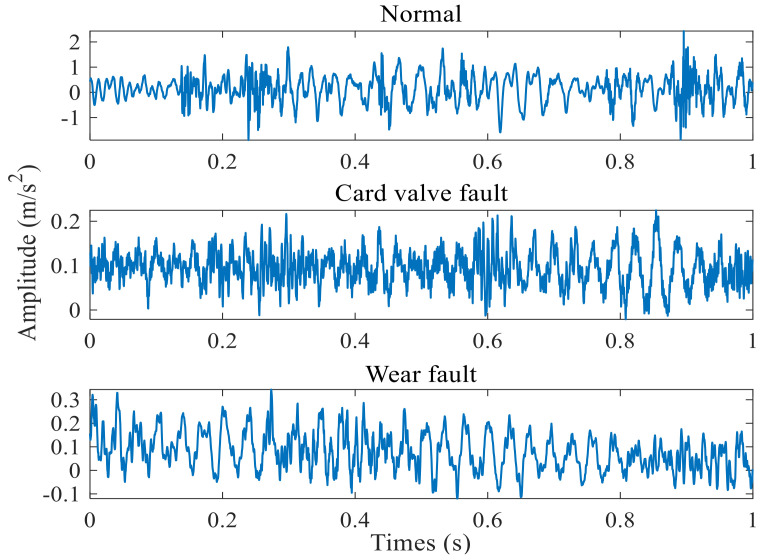
Time−domain waveform of vibration signal of check valve.

**Figure 8 entropy-24-01181-f008:**
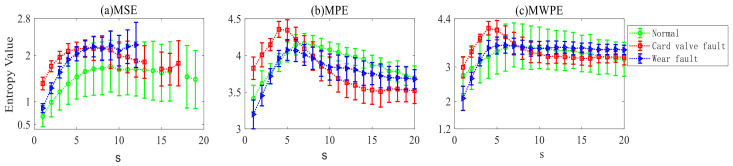
Diagnosis results of MWPE characteristics of check valve under different s values of TELM.

**Figure 9 entropy-24-01181-f009:**
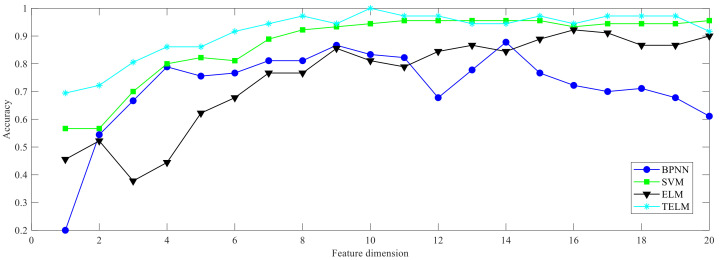
Check valve fault diagnosis accuracy and standardization under different classification models.

**Table 1 entropy-24-01181-t001:** Experimental bearing sample attributes.

Outer Ring (ft)	Inner Ring (ft)	Rotary Element (ft)
0.014	0.07	0.021

**Table 2 entropy-24-01181-t002:** Diagnosis results under different entropy characteristics.

	Diagnostic Time (s)	Accuracy %	Number of Nodes in the Hidden Layer
MSE	0.014	87.50	50
MPE	0.003582	95.833	50
MWPE	0.006596	97.222	50

**Table 3 entropy-24-01181-t003:** Bearing diagnosis results corresponding to the node number of different hidden layers.

	Accuracy under Different Hidden Layer Node Number H							
H	50	80	200	500	800	1000	1500	2000
MSE	55.675%	65.550%	81.111%	93.833%	87.750%	81.111%	95.833%	93.833%
MPE	91.666%	83.333%	79.166%	95.833%	95.833%	91.666%	91.666%	91.666%
MWPE	97.333%	94.444%	97.222%	95.833%	91.666%	95.833%	97.333%	95.833%

**Table 4 entropy-24-01181-t004:** Check valve diagnosis results corresponding to the node number of different hidden layers.

	Accuracy under Different Hidden Layer Node Number H							
H	50	80	200	500	800	1000	1500	2000
MSE	45.675%	68.550%	90.750%	88.550%	91.666%	88.550%	95.833%	95.833%
MPE	91.670%	94.440%	79.166%	91.666%	95.833%	95.833%	91.666%	95.833%
MWPE	97.222%	94.444%	97.222%	95.833%	91.666%	95.833%	97.222%	95.833%

## Data Availability

The data used to support the findings of this study are available from the corresponding author upon request.

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
