# Peer review of "Research on Twin Extreme Learning Fault Diagnosis Method Based on Multi-Scale Weighted Permutation Entropy"

_entropy, 2022, doi:10.3390/e24091181_

Round 1

Reviewer 1 Report

Review of article:  entropy-1834305

 Article: Research on TELM fault diagnosis method based on multi-scale weighted permutation entropy 

 In this work the authors state: “Combining the advantages of TELM, this paper establishes a TELM fault diagnosis model based on Multi-Scale Weighted Permutation Entropy (MWPE). Featuring the characteristic that the vibration signal of the check valve of the high-pressure diaphragm pump changes under non-stable conditions, this paper analyzes the TELM fault diagnosis model based on MWPE, which presents the non-linear dynamic characteristics of the operating check valve. This model expands the applicability of the ELM and improves the accuracy of the fault diagnosis of the model.” 

COMMENTS     

In my view there is no novelty in this work and I need to be convinced about any possible contribution as, in its present form, it is not clear whether it is a straightforward application of existing methods and tools or there is indeed some short of scientific contribution by the authors.

I have a number of comments and questions which need answers, justifications and explanations.

1.     The “contribution” of this work is totally unclear, if any. The authors need to clearly and specifically describe their possible scientific contribution at the end of their introduction.

2.     One major issue with this work is that it discusses a practical problem involving time-series data, without taking into consideration the critical “stationarity” issues involved in time-series and the transformations needed to make them “suitable” for fitting a deep learning model, in order to satisfy the stationarity property. The authors are advised to take a look into some recent literature on the subject (for example: A novel validation framework to enhance deep learning models in time-series forecasting).

3.     The overall structure of this work is not proper. For example, the “related work” or “state of the art” is missing or incompletely presented. The introduction should end by clearly describing the contribution and finally, the contents of following sections.

4.     Section 2 is a mixture of “related works”, and “basic theory” or “background material”.

5.     Section 3, being the proposed “methodology” should be further analyzed and include issues such as the “inspiration” and “expected outcome” of this work.

6.     More details needed on feature extraction issues. Why 60 segments? Why 16 time-domain features, 13 frequency-domain features, 8 wavelet energy features, and 8 wavelet energy entropy features? 

7.     Page 7, line 241: “According to the entropy feature extraction method proposed in this paper”.  The algorithm is not justified (as mentioned in my comment No 5 and 6).  

8.     There is no discussion of the pros and cons of the proposed “methodology”. The authors need to cover this in their Conclusions Section. They also need to discuss future research directions.

9.     The abstract requires re-writing to reflect the problem, the proposed novel or new solution and the major findings.

10.  A link to the code of your algorithm should always be provided for other researchers to reproduce and verify your findings. 

Author Response

Dear editor and reviewers

Thank you very much for your recent email noticing us the required revision of the mentioned paper, which we submitted to the journal ‘Entropy. Thank you for your thoughtful, helpful, and most kind review of manuscript entropy-1834305. We are very grateful to the editor and all the reviewers for their time spending making valuable suggestions and allowing us to submit revisions to the manuscript. We have carefully revised the entire manuscript, and every comment made by the reviewer has been accurately considered and adopted. Respond to the comments point by point below, and point out the modified content. At the same time, the revised text is highlighted in the new manuscript.

We would again like to thank you for your support in advance, and look forward to hearing from you soon.

Kind regards,

Xuyi Yuan

Faculty of Information Engineering and Automation,

Kunming University of Science and Technology,

Kunming, 650500, China

Reviewer 2 Report

The paper is interesting and with direct applicability of modern methods. Please make a general revision of the English and consider the following things to be fixed:

A.      Logical mistakes:

 Line 125 -> Formula 8, not 7 (please check for possible shift of formulas’ numbers)

Line 91 -> ”where is the scale factor is”  (something is missing or appears twice by mistake)

Line 94 – x(i) is missplaced .

Please revise Line 177 (is the number of formula correct?)

60 segments are mentioned at line 212, but in Fig. 2 and at the lines 242,284 there are 60 samples. In my opinnion it is correct to say segments

B.       English:

The statements from the lines 47...49 , respectively 50...52 have to be revised.

Line 246 -> „other parameters s” should be rephrased (e.g. „other values for the parameter s”)

A missing comma (or rephrasing required) at the sentence from the line 307.

C.       Other remarks

Reference [17] mentions a wrong web address

The references are not written according to the template

Lines 172, 180, 195 : ”Where” should be ”where” and the TAB before this word at the beginning of the line should be removed

Table 1 should mention the units (ft)

Author Response

(The authors gave the same response as above.)

Reviewer 3 Report

The manuscript entitled “Research on TELM fault diagnosis method based on multiscale weighted permutation entropy”  presents the application of multiscale weighted permutation entropy (MWPE) method combined with twin extreme learning machine (TELM) method for analysis of vibration signals of a check valve of a high-pressure diaphragm pump. The combination of known method enables to establish  a check valve fault diagnosis model. On my view, such a paper can contribute to knowledge and description of nonlinear and nonstationary characteristics of complex engineering details and there is merit to publish it. However, the current manuscript requires modification and corrections which will enable full understanding of its content. The modifications should be focused particularly onto following issues:

11.        On Page 3, it is written “Calculate the average value of time series … according to Formula 7” However, the average value is calculated from Formula 8.

22.       A reader can find the similar formulation related to Formula 10 on Page 4, Line 147

33.       On Page 5 one can find a sentence “Take the derivative of b  of Equation (1) and set it to 0 …”. However, Equation(1) (why not Formula ... ) does not contain the parameter b. Moreover, the sentence is not mathematically correct.

44.       On Page 5 it is written “The dual problem of the original problem of formulas 7 and 8 is as follows:”. However, one can hardly understand  how to use Formulas (7) and (8) in Formulas (20) and (21).

55       In Figure 2 you are using the abbreviation BPNN. It is necessary to explain it.

66.       In Figures 4 and 8 you are using the parameter s. This parameter you have not defined for the MSE method in Paragraph 2.1.1.

77.       It would be helpful to specify the tested check valve, diaphragm  pump and data acquisition device – types, producers.

88.       In Figures 5 and 9 (not 10) you refer to the SVM and BPNN methods which are not described in the manuscript.

99.       Reference 17 can’ t be find from the reported website. I recommend to use another paper of Loparo Ka dealing with the same subject.

110.   It is not recommended to use abbreviations in paper titles. Please use twin extreme learning method instead of TELM

Author Response

(The authors gave the same response as above.)

Round 2

Reviewer 1 Report

The authors have covered most of my comments to sufficient degree. 

Reviewer 3 Report

I accept changes made by the authors which respond to my comments made to the previous manuscript.  Just, I can recommend to the authors to read carefully the manuscript, particularly inserted and yellow highlighted  parts. E.g. in Abstract the content of sentences in Lines 15-18 has to be checked. These sentences are not clear. The same also holds for Introduction, sentences in Lines 39-41 and in Lines 51-54.